# Biophysical Analysis of Acute and Late Toxicity of Radiotherapy in Gastric Marginal Zone Lymphoma—Impact of Radiation Dose and Planning Target Volume

**DOI:** 10.3390/cancers13061390

**Published:** 2021-03-19

**Authors:** Gabriele Reinartz, Andrea Baehr, Christopher Kittel, Michael Oertel, Uwe Haverkamp, Hans Th. Eich

**Affiliations:** Department of Radiation Oncology, University Hospital Münster, Albert-Schweitzer-Campus 1, 48149 Münster, Germany; andrea.baehr@ukmuenster.de (A.B.); Christopher.Kittel@ukmuenster.de (C.K.); Michael.Oertel@ukmuenster.de (M.O.); Uwe.Haverkamp@ukmuenster.de (U.H.); hans.eich@ukmuenster.de (H.T.E.)

**Keywords:** NTCP, LKB model, side effects, radiation oncology, non-Hodgkin lymphoma, gastric lymphoma

## Abstract

**Simple Summary:**

Prospective evaluation of impact of dose and target volume in radiation planning of gastric lymphoma on organs at risk. New model parameters for calculation of normal tissue complication probabilities were developed from quality-assured cohort data. The study provides practicable data to calculate risks for neighbored organs at risk in modern radiation planning with currently lower radiation doses, representing a basis for future adaptation of previous model parameters.

**Abstract:**

Successful studies on radiation therapy for gastric lymphoma led to a decrease in planning target volume (PTV) and radiation dose with low toxicities, maintaining excellent survival rates. It remains unclear as to which effects are to be expected concerning dose burden on organs at risk (OAR) by decrease in PTV vs. dose and whether a direct impact on toxicity might be expected. We evaluated 72 radiation plans, generated prospectively for a cohort of 18 patients who were treated for indolent gastric lymphoma in our department. As a prospective work, four radiation plans with different radiation doses and target volumes (40 Gy-involved field, 40 Gy-involved site, 30 Gy-involved field, 30 Gy-involved site) were generated for each patient. Mean dose burden on adjacent organs was compared between the planning groups. Cohort toxicity data served to estimate parameters for the Lyman–Kutcher–Burman (LKB) model for normal tissue complication probability (NTCP). These were used to anticipate adverse events for OAR. Literature parameters were used to estimate high-grade toxicities of OAR. Decrease of dose and/or PTV led to median dose reductions between 0.13 and 5.2 Gy, with a significant dose reduction on neighboring organs. Estimated model parameters for liver, spleen, and bowel toxicity were feasible to predict cohort toxicities. NTCP for the endpoints elevated liver enzymes, low platelet count, and diarrhea ranged between 15.9 and 22.8%, 27.6 and 32.4%, and 21.8 and 26.4% for the respective four plan variations. Field and dose reduction highly impact dose burden and NTCP for OAR during stomach radiation. Our estimated LKB model parameters offer a good approximation for low-grade toxicities in abdominal organs with modern radiation techniques.

## 1. Introduction

Indolent lymphomas, particularly of the stomach, are characterized by a low incidence (4% of gastric neoplasias, 8% of non-Hodgkin lymphoma), and therefore the expected number of cases per year is low (1–2 per 100,000). The treatment of gastric lymphoma, particularly for the indolent histologic subtype, is a success story in radiation oncology. In contrast to historic treatment strategies including gastrectomy, an organ-preserving approach has become the state of the art. Patients with *Helicobacter pylori*-negative histology or persisting indolent lymphoma after eradication receive modern radiation therapy with 30 Gy in 1.5-2 Gy single doses by means of the current involved site radiation therapy (ISRT) [1]. With relapse rates of <6% and exceptionally high lymphoma-specific survival of >95%, the outcome for patients is excellent [2]. Patients with gastric marginal zone lymphoma (MZL) are mostly cured, even by applying progressively lower radiation doses and smaller radiation fields.

Over the past decades, trials on indolent lymphomas were conducted in the USA and UK with effective lower radiation doses of 24 or 30 Gy [3,4,5,6,7,8], whereas in Germany, Australia, and in the Netherlands, large studies on gastric indolent lymphoma were continued with doses of a median 40 Gy [9,10,11]. Over the course of three consecutive prospective trials of the German Study Group on Gastrointestinal Lymphoma (DSGL), a stage-adapted target volume concept de-escalated the radiation therapy (RT) from extended field (EF) to involved field (IF). The downsizing of radiation volume led to a reduction of adverse effects, lower disease recurrence, and better survival rates [2]. Low numbers of high-grade toxicities may result in profound benefits. Still, patients experience low-grade toxicities such as diarrhea as acute or chronic—presumably lifelong—side effects that may have a direct influence on their quality of life (QoL) [12].

Dose burden on organs at risk (OAR) is reviewed with the dose volume histogram (DVH) in order to assess effects on normal tissue. Constraints for OAR help to improve the assessment of normal tissue complication probabilities (NTCPs) [13]. Yet, this resource alone bears uncertainties, as markedly different DVHs may have the same NTCP [14]. Calculation of NTCP using the Lyman–Kutcher–Burman (LKB) model offers an additional value in estimating future toxicities by including the volume effects and intrinsic radiosensitivity of organs [14,15,16,17]. With the modern radiation concept of ISRT with 30 Gy for gastric lymphoma, as conventional treatment outside of studies, it is expectable that dose constraints from the “Analyses of Normal Tissue Effects in the Clinic” (QUANTEC) data will not be exceeded for organs in the abdomen. These data offer predictions for high-grade toxicities, as for example the endpoint radiation-induced liver disease (RILD), which might be expected for mean doses higher than 28 Gy [18]. Even though the original paper describes which low-grade toxicities might arise, their endpoint represents a high-grade toxicity syndrome. Similarly, subsequent publications with further development of LKB model parameters concerning liver toxicities focused on RILD [19,20]. Analogously, the recommendations for other gastrointestinal (GI) organs such as the duodenum or bowel refer to grade 3 or higher toxicities [21], and endpoints for LKB model adjustments were, e.g., severe gastric or duodenal ulceration with bleeding [22,23].

Considering the low frequency of high-grade side effects on the one hand and the influence of common low-grade side effects on QoL on the other hand, a detailed examination of NTCP, even for low-grade toxicity, is of high interest for oncologists.

### Purpose

The aim of this study was the first-time application of the LKB model for assessing the NTCP in radiation therapy of gastric lymphoma and its adaptation to current low radiation doses. We compared different concepts of radiation treatment using newly created treatment plans for a cohort of patients. The radiation dose exposure on OAR was assessed together with the NTCP values. Coefficients for the LKB model were calculated with quality-assured data, which can be used for the evaluation of low-grade adverse reactions such as outcome variables of diarrhea, elevation of transaminases, and thrombocytopenia.

The implementation of the LKB model in radiation treatment of gastric lymphoma provides the option of a standardized assessment of NTCP and thereby the comparison between modern radiation concepts.

## 2. Materials and Methods

Records of patients who were treated for gastric lymphoma in 2005–2017 were screened for treatment and follow-up reports. Inclusion criteria were gastric marginal zone lymphoma, stage I–II, treated with 40 or 39.6 Gy without concurrent chemotherapy. A total of 18 patients were eligible. Intensity modulated radiation techniques (IMRT) were used for the original plans. The CT (computed tomography) scans were performed on an empty stomach after a fast of at least 4 h or overnight, with patients in supine position with arms up using a customized immobilization device. A small volume (<50 mL) of oral contrast medium was used in all cases, and intravenous contrast medium was recommended if there were suggestive lymph nodes. Respiratory motion was assessed with fluoroscopy or 4D CT.

Toxicities were noted according to the “common toxicity criteria of adverse events” (CTCAE) [24]. The interval to detect early toxicity ranged from the therapy start to 90 days after radiotherapy; toxicities occurring later than 90 days after radiation treatment were defined as late toxicities. Follow-ups were conducted until 1 August 2020.

### 2.1. Planning

To evaluate impact of planning target volume and radiation dose, on the basis of the original CT scans, we performed modern IMRT planning: as 40 Gy IF, 40 Gy involved site (IS), 30 Gy IF, and 30 Gy IS. Abdominal OAR and planning target volume (PTV) for IF and IS were defined and outlined. International Lymphoma Radiation Oncology Group (ILROG) guidelines were used as a reference [1]. All plans were generated for a linear accelerator using 6 MeV photons. The prescription dose was 15 fractions of 2 Gy (30 Gy total dose) or 20 fractions of 2 Gy (40 Gy total dose) to a mean dose of 100% in the PTV. Coverage of at least 95% of the PTV was attempted. All plans were created with individual optimization.

### 2.2. Evaluation of Dose Burden for OAR

Mean doses for abdominal organs as well as the volume receiving ≥35 Gy (V35) for small bowel volume and V30 for liver volume from DVHs were compared between the 4 groups: (1) 40 Gy IF, (2) 40 Gy IS, (3) 30 Gy IF, and (4) 30 Gy IS. As cytopenia and GI side effects are common toxicities, we also evaluated mean doses on the spleen, duodenum, and small bowel. The impact of target volume decrease was investigated by comparing group (1) and (2) as well as group (3) and (4). The impact of radiation dose decrease on dose uptake of OAR was measured by comparing group (1) and (3) as well as group (2) and (4). Pair-wise Wilcoxon tests were used for statistical analyses of median values; *p*-value of <0.05 was regarded as significant.

### 2.3. Estimation of NTCP

The LKB model is described by 3 parameters [15,16]: the estimated dose on OAR where expected toxicity rate is 50% (TD50); m, which is a measure for the standard deviation of TD50 and represents the steepness of the NTCP curve; and n, which introduces the volume effect of the OAR into the model.
(1)$$NTCP=\;\frac{1}{\sqrt{2\cdot \pi }}\int_{\infty }^{t}e\frac{-t^{2}}{2}dt$$

*t* is composed of the m, TD50, and *D_max_*, which is comparable to the maximum dose in the DVH, thus showing their interdependencies:(2)$$t=\frac{D_{max}-TD_{50}\left(V\right)}{m\cdot TD_{50}\left(V\right)}\;$$

TD50 is made up on the irradiated partial volume Veff and is described by
(3)$$TD_{50}\left(V\right)=TD_{50}\cdot V_{eff}^{-n}$$

The effective volume method transforms the DVH into a single-stage (δ-shaped) DVH with a high effective volume Veff, with the following result
(4)Veff=∑i=1j[ (diDmax)1m·vi]

### 2.4. Here, di and vi Were Taken from the Differential DVH of the Patient

In order to generalize the observed side effects (grade 1 or higher), we optimized elevated transaminases (liver), diarrhea (small bowel), and low platelet count (spleen) and the resulting probability, n, m, and TD50 for the patient cohort by the use of the Levenberg–Marquardt algorithm (LMA) [14], starting from the basic values from the literature [20,25].

### 2.5. Estimation of NTCPs

NTCPs for each radiation plan (group 1–4) were calculated with the LKB model as described above using the new estimated values for TD50, m, and n for toxicities of the liver, bowel, and spleen. The impact of target size and radiation dose was estimated using a mixed-model regression. A *p*-value of <0.05 was regarded as significant. The correlations between OAR dose and NTCP were estimated using Pearson’s coefficient.

### 2.6. Estimation of Dose NTCPs Using Data from the Literature

We calculated the risks for high-grade toxicities, i.e., RILD and bowel perforation/obstruction. Each risk was calculated for all 4 groups of the radiation plans. For RILD values, those described by Dawson et al. [20] were used (m 0.12, n 0.97, TD50 45.8 Gy), and for bowel perforation/obstruction, values from Burman et al. [25] were used (m 0.16, n 0.15, TD 50 55 Gy).

## 3. Results

### 3.1. Patients’ Treatment and Outcome

A total of 26 patients were treated for gastric marginal zone lymphoma stage I-II in 2005–2017. In total, 19 received 39.6 or 40 Gy as maximum dose, and 1 was excluded because of a simultaneous chemotherapy. Characteristics of the remaining participants and their respective original treatments are shown in Table 1.

Median age was 62.4 years. At diagnosis, 6 patients showed *Helicobacter pylori* infection, whereas 12 patients primarily were *Helicobacter pylori*-negative. Apart from eradication therapy, no other previous therapies were applied. At the beginning of radiation therapy, all of the 18 patients were *Helicobacter pylori*-negative, while lymphoma was persisting for at least 12 months after eradication. The endoscopic description before starting the radiation in the majority of patients showed ulcerative mucosal changes as typical pathological finding of the lymphoma, and only two patients had no adverse mucosal findings.

Most patients received IF radiation. Mean follow-up was 81.6 months. Two patients died since the end of therapy, whereas none showed lymphoma progress. The endoscopic follow-up after radiation treatment had no reports of recurrences, while in half of the patients there was a remaining discrete scar or slightly reddened zone at the lymphoma site concerned. Most common acute toxicities were nausea and fatigue grade 1–2, and most common late toxicity was elevated liver transaminases grade 1–2 (Table 2). One patient who had been diagnosed with myelodysplastic syndrome before radiation treatment showed low counts of blood cells at grade 3. One patient had a self-limiting GI bleeding for four years after radiation. No grade 4 toxicities were reported.

### 3.2. Dose Exposure According to Radiation Planning

Figure 1 shows the median mean doses for abdominal OAR as well as median V30 for liver and V35 for small bowel as calculated for the four groups of prescription.

Comparison of 40 Gy IF vs. 40 Gy IS showed significantly decreased doses on five OAR: duodenum (median 38.9 vs. 21.7 Gy *p* < 0.001), right kidney (10.1 vs. 6.9 Gy *p* < 0.001), left kidney (9.3 vs. 6.1 Gy *p* < 0.001), mean liver (18.1 vs. 16.7 *p* < 0.001), liver V30 (14.5 vs. 12.7% *p* < 0.001), small bowel mean (9.8 vs. 6.5 Gy *p* < 0.001), and V35 (9.15 vs. 8.3 Gy *p* < 0.001). Median dose reduction was 2.3 Gy.

Comparison of 30 Gy IF vs. 30 Gy IS showed dose decrease for six organs: duodenum (median 29.8 vs. 16.2 Gy *p* < 0.001), spleen (17.4 vs. 17.1 Gy *p* = 0.014), right kidney (7.6 vs. 6.2 Gy *p* < 0.001), left kidney (6.9 vs. 4.6 Gy *p* < 0.001), liver mean (13.7 vs. 12.5 Gy *p* < 0.001), liver V30 (3.2 vs. 3% *p* < 0.001), and small bowel mean (6.6 vs. 6.1 Gy *p* < 0.001). Median dose reduction was 1.7 Gy.

Comparison of 40 Gy IF vs. 30 Gy IF revealed significant differences for six OAR: duodenum (median 38.9 vs. 29.8 Gy *p* < 0.001), spleen (23.3 vs. 17.4 Gy *p* < 0.001), right kidney (10.1 vs. 7.6 Gy *p* < 0.001), left kidney (9.3 vs. 6.9 Gy *p* < 0.001), liver mean (18.1 vs. 13.7 Gy *p* < 0.001), liver V30 (14.5 vs. 3.2% *p*< 0.001), small bowel mean (9.8 vs. 6.6 Gy *p* < 0.001), and V35 (9.15 vs. 0% *p* < 0.001). Median dose reduction was 3.1 Gy.

Comparison of 40 Gy IS vs. 30 Gy IS showed significant differences for six organs: duodenum mean (median 21.7 vs. 16.2 Gy *p* < 0.001), spleen (22.8 vs. 17.1 Gy *p* < 0.001), right kidney (6.9 vs. 6.2 Gy *p* < 0.001), left kidney (6.1 vs. 4.6 Gy *p* < 0.001), liver mean (16.7 vs. 12.5 Gy *p* < 0.001), liver V30 (12.7 vs. 3% *p* < 0.001), small bowel mean (8.2 vs. 6.1 Gy *p* < 0.001), and V35 (8.3 vs. 0% *p* < 0.001). Median dose reduction was 3.2 Gy.

Comparison of 40 Gy IS vs. 30 Gy IF revealed significantly lower mean doses for three organs: spleen (median 22.8 vs. 17.4 Gy *p* < 0.001), liver mean (16.7 vs. 13.7 Gy *p* < 0.001), liver V30 (12.7 vs. 3.2% *p* < 0.001), small bowel mean (8.2 vs. 6.6 Gy *p* = 0.014), and V35 (8.9 vs. 0% *p* < 0.001). Median dose reduction on OARs was 0.13 Gy.

Comparison of 40 Gy IF vs. 30 Gy IS displayed significantly lower mean doses for all OAR: duodenum mean (median 38.9 vs. 16.2 Gy), spleen (23.3 vs. 17.4 Gy), right kidney (10.1 vs. 6.2 Gy), left kidney (9.3 vs. 4.6 Gy), liver V30 (14.5 vs. 3.2%), small bowel mean (9.8 vs. 6.1 Gy), and V35 (9.15 vs. 0%), each with *p* < 0.001, and also liver mean (18.1 vs. 12.5 Gy *p* < 0.035). Median dose reduction on OARs was 5.2 Gy.

### 3.3. Estimation of NTCPs

The toxicities from our patient collective were used for estimation of parameters for toxicities according to the LKB model. Details are described in the methods section. Results are shown in Table 3, which also shows the values for RILD and bowel perforation found in literature.

Probabilities for toxicities as indexed in Table 3 were calculated for all 18 patients and as median values for the four planning groups. Figure 2 shows the results.

NTCPs for elevated liver transaminases were 22.8%, 21.7%, 16.6%, and 15.9% for the four groups. Comparison of groups as performed for dose burden on OAR revealed significant differences for all comparisons (group 1 vs. 2/group 3 vs. 4/group 1 vs. 3/group 2 vs. 4/group 2 vs. 3/group 1 vs. 4, each with *p* < 0.001). A stronger correlation for V30 vs. mean dose with NTCP value was observed (Pearson’s coefficient = 0.97 vs. 0.8). Table 4 shows cohort toxicity rates vs. estimated rates for 40 Gy IF radiation.

NTCPs for low platelet count were 32.4%, 33.6%, 29.8%, and 27.6%. Significant decreases were seen for comparing group 3 vs. 4 (30 Gy IF vs. 30 Gy IS, *p* = 0.01), group 2 vs. 4 (40 Gy IS vs. 30 Gy IS, *p* = 0.02), group 2 vs. 3 (40 Gy IS vs. 30 Gy IF, *p* = 0.025), and group 1 vs. 4 (*p* = 0.035). Pearson’s coefficient comparing NTCP value and mean dose on spleen was 0.91.

NTCPs for diarrhea were 26.4%, 26.1%, 21.5%, and 21.8% for the four groups. Comparison revealed significant differences for all comparisons (each with *p* < 0.001). Pearson’s coefficient revealed a stronger correlation for NTCP with V35 vs. mean small bowel dose (0.95 vs. 0.54).

Linear regression modelling showed significant impact of dose but not of target volume decrease on estimated NTCP (each with *p* < 0.001). The strongest impact was estimated in the case of NTCP spleen (regression coefficient of 6.4).

### 3.4. Estimation of High-Grade Toxicities Using Data from the Literature

Calculation of the risks for RILD and ulceration/perforation of the small bowel for our cohort were performed. Thus far, no data about LKB calculations are available for spleen toxicity. Table 5 shows our results. RILD probability was 0% for all groups, while bowel toxicity probability was 0% in 30 Gy radiation and 0.06%/0.04% in 40 Gy IF/IS radiation groups.

## 4. Discussion

This study describes a cohort of patients to compare different sizes of planning target volumes and radiation doses regarding their impact on dose burden and NTCPs of OAR. Epidemiologic data of our patients are in accordance with populations of gastric lymphoma patients from the literature [2], and survival data also represents typically favorable high rates of 5-year overall and lymphoma-specific survival [11,26,27]. Toxicities are similarly comparable [2,28]—nausea and diarrhea as early and increase of transaminases as late toxicity were very common, albeit hematopoietic side effects were rare. This might have been due to the fact that we excluded patients who received systemic therapies for the MZL, which would increase impact on hemotopoiesis. Nevertheless, low platelet count was reported as acute and late toxicity and optimization of the LKB model parameters for this specific endpoint as a function of the spleen DVH was possible.

A combined consideration of DVH and NTCP calculation as well as clinical factors and patients’ preferences are important to provide a safe and modern therapy [14]. Despite increasing accuracy of NTCP estimation, the method still bears the weakness of relying mainly on one CT dataset for each patient, which served as a base for radiation planning. In modern approaches, the inaccuracy derived from depending on breath or other motion of organs including the stomach needs to be considered [29]. Studies showed a significant impact of motion on mean doses and respective NTCP for liver [30,31]. Similarly, in a study by McCulloch et al., NTCP differed between planned and accumulated dose for the stomach and duodenum [30]. This effect grew with higher dosages, and for doses ≥20 Gy, NTCP models derived from planning dose vs. accumulated dose even underestimated toxicity rates.

Published studies on NTCP for other organs at risk after radiation treatment comprise various patient numbers and provide different informative values. Other authors using a similar approach by generating comparable radiation plans with consistent radiation dosage reported similar limited numbers of patients as involved in this study [30]. Another publication, including a larger number of patients, merged patients from different studies to optimize existing NTCP parameters, but bore limitations, e.g., of slightly to noticeably varying radiation planning between the included studies and usage of vague data on doses for organs at risk [23]. Results of our study encourage further medical physics research of the Lyman–Kutcher–Burman (LKB) model parameters as an additional value concerning future toxicities, particularly for currently lower radiation doses.

Multiple studies use the LKB model for estimation of liver or GI toxicities, yet few report about probabilities for lower-grade toxicity [23]. In our cohort, only a minority of patients suffered from severe side effects but most of them experienced early grade 1–2 toxicities with GI side effects. Decrease of planning target volume and dose leads to significantly decreased mean doses for 3–7 of the considered OAR, whereas median dose reduction differs between 0.13–5.3 Gy. As high correlation coefficients indicate, the mean doses are good indicators for NTCP as calculated with the LKB model.

Mean liver dose ranged from 12.5 (30 Gy IS) to 18.2 Gy (40 Gy IF), and V30 ranged from 3% (30 Gy IS) to 14.5% (40 Gy IF). Significant improvements throughout field or dose decreases were observed for these two dose values as well as NTCP for the endpoint elevated transaminases (15.9% (30 Gy IS)–22.8% (40 Gy IF)). This might be of special interest for treatment of patients with comorbidities concerning the liver such as cirrhosis or hepatitis infections [19], as these seem to possess markedly different radiation sensitivity. Special attention needs to be paid to the findings concerning intra-fractional organ motion as described above [31]. As our cohort data were derived from patients who received different field sizes and techniques, the observed rate of liver toxicity lay discretely above the estimated value for 40 Gy IF, but still presented a close approximation (27.7% vs. 22.8%). Overall probabilities for radiation-induced liver disease and bowel perforation/obstruction were neglectable.

As GI toxicities in particular might pose a heavy burden [32], while they are also common side effects in abdominal radiation therapy [23,33], reducing these toxicities with modern concepts is very promising. Mean doses on small bowel ranged from 6.8 (30 Gy IS) to 9.1 Gy (40 Gy IF) and were significantly reduced by lowering of dose and planning target volume. As the mean dose on this serial organ might be of little importance, a threshold dose (V35), which ranged from 0 (30 Gy IS/IF) to 9.15% (40 Gy IF) was evaluated. Estimated NTCP (diarrhea) declined significantly with decreased doses and target volume. Our parameters offer a good approximation for diarrhea as indicated by low divergence between estimated and reported toxicity rates (22.2 vs. 26.4%, 40 Gy IF). With upcoming low-dose radiation concepts for lymphoma, QUANTEC [21] recommendations to not exceed 45 Gy for ≥120 ccm of the bowel is of little help, and thus future investigations should focus on developing new thresholds for doses ≤30 Gy. Within this study, the model parameters for the biophysical calculation were adapted to the cohort and have been revised. Hereafter, an adjustment of the past model parameters should be considered by further analysis in a larger cohort of patients.

Inhibition of splenic function after radiation for lymphoma is a well-known phenomenon [34,35,36]. This effect is used for patients suffering from symptomatic splenomegaly [37]. Several biological aspects such as, e.g., aberration in cytokine release might be of importance [38], making radiation effects of the spleen an interesting factor, even when effects of radiation combined with checkpoint-inhibitors are discussed. Distinct loss of splenic function might not be of importance in mean doses under 10 Gy [39]. However, a study by Liu et al. showed that the mean dose and V5 of the spleen can lead to a decrease in absolute lymphocyte count in patients receiving radiation [40]. Our investigation now addresses the impact of the splenic dose on platelet count. Mean doses ranged from 17.1 Gy (30 Gy IS) to 23.3 Gy (40 Gy IF). Significantly reduced mean dose was reached by field and dose decrease for all comparisons, with the exception of 40 Gy IF vs. 40 Gy IS. The NTCP of decreased thrombocytes was significantly lower for 30 Gy IS vs. all other groups of concepts and similarly for 30 Gy IF vs. 40 Gy IS. The reported toxicity rate (33%) was very close to the estimated value of 32%. As in large prospective trials, hematotoxic side effects were predominant; this effect is of high interest for patients and upcoming studies such as the ongoing multicenter trial “GDL-ISRT-20 Gy” (NCT 04097067).

A limitation of this study is the description of toxicities according to paper-based documentation, which substantially depends on the accurateness of the physician. Unlike for retrospective evaluation of the greater lymphoma studies, it was not possible to evaluate the important point of subjective severity of side effects as it is for example possible via use of the SOMA scale [41], although at least for nausea, the CTCAE criteria mainly relies on subjective need for antiemetic therapy.

The decrease in mean dose burden on OAR comparing 40 Gy IF vs. 30 Gy IS was the strongest decline in our comparison but still only showed a difference of 5.3 Gy. This shall be a point of discussion as this small difference might not be of clinical importance. Yet, in large studies, toxicity rates dropped with reduced field sizes, indicating a clinical benefit even with small dose differences [2]. A significant impact between hematotoxicity and low-dose exposition on splenic tissue was reported [40], and a dose effect especially for low doses seems to exist in small-bowel radiation [21,42]. This underlines the importance of even little mean dose differences, as seen when comparing 30 Gy IF vs. 30 Gy IS (bowel median 6.6 vs. 6.1 Gy), even though evaluation with a regression model offered no significant impact of field size decrease on calculated NTCPs. Nevertheless, impact of toxicities on QoL for cancer survivors deserves more attention, and future studies should evaluate the effects of dose decrease on long-term symptoms.

## 5. Conclusions

For the high curative purpose of treatment, the potential toxicity of radiation treatment of gastric indolent lymphoma is of rising importance. Decrease of dose and target volume in modern radiation concepts leads to a significantly lower dose burden on adjacent abdominal organs. In contrast, the models and their parameters for calculating normal tissue complication probability (NTCP) of radiotherapy are antiquated, particularly with regard to lymphoma, wherein they need to be adjusted to the nowadays lower radiation doses.

Estimated and reported toxicity rates for severe side effects are low. Cohort data-based estimation of new LKB model parameters for NTCP offers practicable data to calculate risks for neighbored organs at risk during radiation planning.

The study provides deeper insights into the influence of dose versus field decrease in abdominal radiation and for the first time develops new LKB model parameters for normal tissue complication probability in planning of modern low-dose radiation treatment of gastric lymphoma. These findings could be a basis for future adaptation of the model parameters.

## Figures and Tables

**Figure 1 cancers-13-01390-f001:**
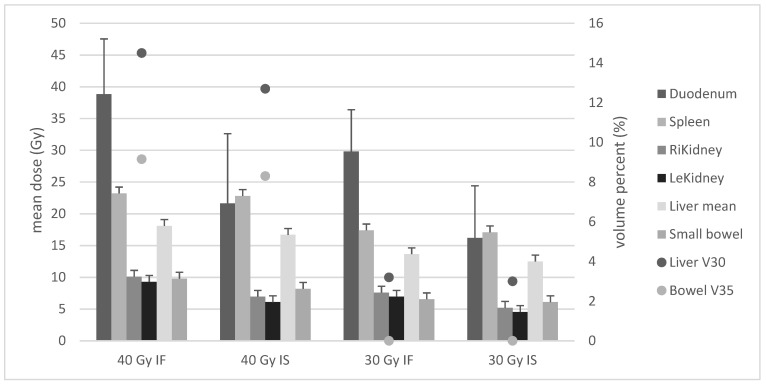
Bars show mean doses for abdominal organs at risk (OAR) as calculated for the four groups. Medians of mean doses are indicated in Gy on the left scale. Dots show the V30 for liver and V35 for small bowel. Median volume percentages are indicated on the right scale. OAR = organs at risk, LeKidney = left kidney, RiKidney = right kidney.

**Figure 2 cancers-13-01390-f002:**
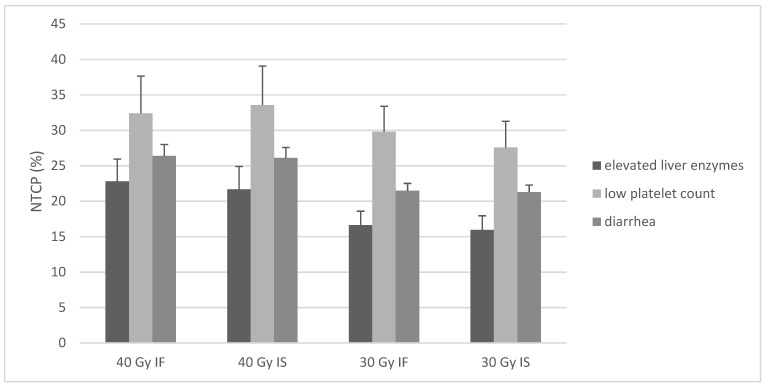
Median NTCPs (%) and standard deviation as calculated with the LKB model using data from the cohort of 18 patients, their CT = computed tomography, and follow-up data on toxicities.

**Table 1 cancers-13-01390-t001:** Patient cohort data.

**Sex, Number of Patients (*n*)**	
Female	6
Male	12
**Age (Years)**	
Median	62.4
Range	31–85
**Stage**	
I	9
II_1_	9
II_2_	0
**PTV Definition (Original Treatment)**	
Extended field (abdomen)	1
Extended field short (abdomen without pelvis)	5
Involved field	10
Involved site	2
**Follow-Up (Months)**	
Mean	81.6
Range	13.5–182
Patients deceased (*n*)	2 (11%)
Overall survival	89%
Lymphoma relapse (*n*)	0 (0%)
Mean lymphoma-free survival (months)	81.6
Five-year-overall survival	94%
Five-year lymphoma-specific survival	100%

Data of the 18 included patients showing information on demographic data, original radiation field, and survival parameters. Two patients died during follow-up time, and no patient showed lymphoma progress. *n* = number, PTV = planning target volume.

**Table 2 cancers-13-01390-t002:** Cohort toxicities.

Early Toxicities, Grade 1–2 (*n*)	
Gastrointestinal Tract	
Nausea	13 (72%)
Emesis	2 (11%)
Constipation	3 (16.7%)
Diarrhea	3 (16.7%)
Abdominal pain	1 (5.5%)
Hematopoietic system	
Anemia	2 (11%)
Low lymphocyte count	2 (11%)
Low platelet count	4 (22%)
Urinary tract	
Increased urinary frequency	1 (5.5%)
Acute infection	1 (5.5%)
Other	
Fatigue	6 (33.3%)
Hypokalemia	1 (5.5%)
Early Toxicities, Grade 3–4 (*n*)	
Hematopoietic system	
Low platelet count	1 (5.5%)
Low lymphocyte count	1 (5.5%)
Late Toxicities Grade 1–2 (*n*)	
Hepatobiliary system	
Elevated transaminases	5 (27.7%)
Hematopoietic system	
Low platelet count	1 (5.5%)
Low lymphocyte count	2 (11%)
Gastrointestinal tract	
Diarrhea	1 (5.5%)
Heartburn	2 (11%)
Constipation	2 (11%)
Loss of appetite	1 (5.5%)
other	
Fatigue	2 (11%)
Late Toxicities, Grade 3–4 (*n*)	
Gastrointestinal tract	
Gastric bleeding	1 (5.5%)

Toxicities that occurred during radiation treatment and follow-up. Toxicities occurring > 90 days after radiation are considered as late toxicities. Grading according to the common criteria of adverse events version 5.0. Grade 3 early and late hematopoietic abnormalities were reported for a patient with a myelodysplastic syndrome, which was known before radiation treatment. One patient showed a self-limiting gastric bleeding four years after treatment. *n* = number.

**Table 3 cancers-13-01390-t003:** Estimated Lyman–Kutcher–Burman (LKB) model parameters for low-grade toxicities and respective literature values for liver and bowel toxicities.

Organ	Endpoint	*n*	m	TD50 (Gy)
Spleen	low platelet count	0.5	0.85	35.0
Small bowel	diarrhea	0.15	0.79	55.0
*Small bowel* [25]	*perforation/obstruction*	*0.15*	*0.16*	*55.0*
Liver	elevated transaminases	0.32	0.61	39.6
*Liver* [20]	*RILD*	*0.97*	*0.12*	*45.8*

Results of calculation for m, n, and TD50 for low platelet count, diarrhea, and elevated liver transaminases. Italic lines show literature values for bowel and liver high-grade toxicities. TD50 = estimated dose on OAR where expected toxicity rate is 50%, m = a measure for the standard deviation of TD50 representing the steepness of the normal tissue complication probability (NTCP) curve, *n* = introduces the volume effect of the OAR into the model.

**Table 4 cancers-13-01390-t004:** Cohort toxicity rates vs. estimated NTCP.

Endpoint	Cohort Toxicity Rate	Estimated NTCP
Low platelet count	33%	32% (SD 5.25)
Diarrhea	22.2%	26.4% (SD 1.57)
Elevated transaminases	27.7%	22.8% (SD 3.15)

The middle column indicates the toxicity rate as found in the patient record data. The right column shows the estimated NTCP as calculated for 40 Gy involved field (IF). SD = standard deviation.

**Table 5 cancers-13-01390-t005:** Estimated high-grade toxicities.

Organ at Risk (OAR)	Endpoint	IF 40 Gy	IS 40 Gy	IF 30 Gy	IS 30 Gy
Small bowel	ulceration/perforation	0.06%	0.04%	0%	0%
Liver	RILD	0%	0%	0%	0%

Estimated NTCP for two OAR and the respective endpoints as calculated from values for TD50, m, and n found in the literature [20,25]. TD50 = estimated dose on OAR where expected toxicity rate is 50%, m = a measure for the standard deviation of TD50 representing the steepness of the NTCP curve, *n* = introduces the volume effect of the OAR into the model.

## Data Availability

The data presented in this study are available in the article.

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
