# Peer review of "Biophysical Analysis of Acute and Late Toxicity of Radiotherapy in Gastric Marginal Zone Lymphoma—Impact of Radiation Dose and Planning Target Volume"

_cancers, 2021, doi:10.3390/cancers13061390_

Round 1
Reviewer 1 Report
The authors report the results of a prospective trial evaluating the impact of 4 different protocols (according to the dosage and the field) of radiation therapy in gastric marginal lymphoma. They measured the mean dose burden and the adjacent organ toxicity accross the 4 arms, based on a specific model (KLB). They conclude that an involved site radiation at a 30GY dose is less toxic and equivalent in terms of disease control as compared to higher dose and more extended field.
If the scientific idea is pertinent and the method interesting, the very low number of patients (n=18) makes the conclusion hazardous. The results depicted in figure 2 are not convincing. The number of patients treated in each arm does not allow to draw definitive conclusions as it is concluded in this ms.
More data are needed to describe the patients: previous therapy, endoscopic description with follow-up, incidence of HPylori....
Author Response
Response to Reviewer 1 Comments
Dear reviewer,
thank you for your reviewing of this manuscript and your opinion on our paper.
Point 1: If the scientific idea is pertinent and the method interesting, the very low number of patients (n=18) makes the conclusion hazardous. The results depicted in figure 2 are not convincing. The number of patients treated in each arm does not allow to draw definitive conclusions as it is concluded in this ms.
Response 1:
We understand the reviewer’s thoughts concerning the number of involved patients. According to the reviewer’s comment we have added a detailed description of the involved lymphoma entity, of its prognosis and the prospective work of the study to the introduction-section as well as to the abstract and the conclusions-section:
Indolent lymphomas, in particular of the stomach, are characterized by a low incidence (4 % of gastric neoplasias, 8 % of non-Hodgkin lymphoma), therefore the expected number of cases per year is low (1-2 per 100000). Their prognosis is excellent, patients with gastric marginal zone lymphoma are mostly cured, even by applying progressively lower radiation doses and smaller radiation fields.
For this study as a prospective work, 72 radiation plans with different radiation doses and target volumes (40 Gy IF, 40 Gy IS, 30 Gy IF, 30 Gy IS) were generated for the cohort of 18 patients.
For the high curative purpose of treatment, the potential toxicity of radiation treatment of gastric indolent lymphoma is of rising importance. In contrast, the models and their parameters for calculating normal tissue complication probability (NTCP) of radiotherapy are antiquated, particularly in regard to lymphoma they need to be adjusted to the nowadays lower radiation doses.
This study for the first time develops new LKB model parameters for normal tissue complication probability in planning of low dose radiation of gastric lymphoma. The findings of this study can be a basis for prospective adaptation of the LKB model parameters, in particular for modern low-dose radiation treatment of gastric lymphoma.
First of all, the results depicted in figure 2 allow limited conclusions due to the involved number of n=18 patients. But since this study involves a rare disease, the depicted 3 parameters showing significant differences in all compared subgroups of the 72 radiation plans represent pioneering results. According to the reviewer’s comment we added comparable studies of other authors with similar numbers of patients in the discussion-section:
Published studies on NTCP for other organs at risk after radiation treatment comprise various numbers of patients and provide different informative value. Other authors using a similar approach by generating comparable radiation plans with consistent radiation dosage reported about similar limited numbers of patients as in this study[30]. Another publication, including a larger number of patients, merged patients from different studies to optimize existing NTCP parameters, but bears limitations e.g. of slightly to noticeably varying radiation planning between the included studies and usage of vague data on doses for organs at risk[23]. Results of our study encourage for further medical physics research of the Lyman-Kutcher-Burman (LKB) model parameters as an additional value concerning future toxicities, in particular for currently lower radiation doses.
Point 2: More data are needed to describe the patients: previous therapy, endoscopic description with follow-up, incidence of HPylori....
Response 2:
According to the reviewer’s comment we added the following descriptions of previous therapy, endoscopic description with follow-up, incidence of helicobacter pylori infection in the results-section:
At diagnosis 6 patients showed helicobacter pylori infection whereas 12 patients primarily were helicobacter pylori negative. Apart from eradication therapy no other previous therapies were applied. At the beginning of radiation therapy all of the 18 patients were helicobacter pylori negative while lymphoma was persisting for at least 12 months after eradication. The endoscopic description before starting the radiation in the majority of patients showed ulcerative mucosal changes as typical pathological finding of the lymphoma, only 2 patients had no adverse mucosal findings. The endoscopic follow-up after radiation treatment had no reports of recurrences, in half of the patients remained a discrete scar or slightly reddened zone at the lymphoma site concerned.
-----------------------------------------------------------------------------------------------------------------
Responses relating to general Reviewer 1 questions:
These issues were also addressed by Reviewer 3.
Question: Does the introduction provide sufficient background and include all relevant references?
Response: We understand the reviewer’s thought on the introduction-section. We hereby note that the introduction is clearly structured and summarizes different dose and target volume concepts in gastric marginal zone lymphoma and the aim of applying the model of Lyman-Kutcher-Burman (LKB). According to the reviewer’s comment we have added a detailed description of the involved lymphoma entity, of its prognosis and the prospective work of the study in the introduction-section. This issue is also addressed in our Response 1 above.
Question: Is the research design appropriate?
Response: We thank the reviewer for this comment and hereby note that the LKB model provides additional benefit for estimation of future toxicities, by including volume effects and intrinsic radiosensitivity of organs, allowing a better assessment of low grade toxicity in abdominal organs. This issue is also addressed in our Response 1 above.
Question: Are the methods adequately described?
Response: According to the reviewer’s comment more details of the methods were added in the ‘Materials and Methods’-section):
Intensity Modulation Radiation Techniques (IMRT) were used for the original plans. The CT scans were performed with an empty stomach after a fast of at least 4 hours or overnight, patients in supine position with arms up using customized immobilization device.
A small volume (<50 ml) of oral contrast medium was used in all cases, intravenous contrast medium was recommended if there were suggestive lymph nodes. Respiratory motion has been assessed with fluoroscopy or 4D CT.
The interval definition for early toxicity was added in the ‘Materials and Methods’-section:
The interval to detect early toxicity ranged from the therapy start to 90 days after radiotherapy, toxicities occurring later than 90 days after radiation treatment were defined as late toxicities.
And in the following part of the ‘Materials and Methods’-section we described the planning process, the evaluation of radiation dose burden and explained the NTCP calculation.
Question: Are the results clearly presented?
Response: According to the reviewer’s comment we described very detailed the results with regard to clinical results and to plan-compairing results. We added more data to describe the patients in the results-section:
At diagnosis 6 patients showed helicobacter pylori infection whereas 12 patients primarily were helicobacter pylori negative. Apart from eradication therapy no other previous therapies were applied. At the beginning of radiation therapy all of the 18 patients were helicobacter pylori negative while lymphoma was persisting for at least 12 months after eradication. The endoscopic description before starting the radiation in the majority of patients showed ulcerative mucosal changes as typical pathological finding of the lymphoma, only 2 patients had no adverse mucosal findings. The endoscopic follow-up after radiation treatment had no reports of recurrences, in half of the patients remained a discrete scar or slightly reddened zone at the lymphoma site concerned. This issue is also addressed in our Response 2 above.
We compared and described median doses in all groups and calculated the NTCPs.
To simplify the Table 3 and Table 5 for the reader the three parameters of the LKB model „TD50“, „m“ and „n“ are now described in the footnotes below Table 3 and below Table 5, as suggested.
Question: Are the conclusions supported by the results?
Response: Thank you for your review. We hereby refer to our description of the results and the discussion of the limitations of the study in the discussion-section.
First of all, the results allow limited conclusions due to the involved number of n=18 patients. But since this study involves a rare disease, the depicted 3 parameters showing significant differences in all compared subgroups of the 72 radiation plans represent pioneering results.
According to the reviewer’s comment we added comparable studies of other authors with similar numbers of patients in the discussion-section:
Published studies on NTCP for other organs at risk after radiation treatment comprise various patient numbers and provide different informative value. Other authors using a similar approach by generating comparable radiation plans with consistent radiation dosage reported about similar limited numbers of patients as involved in this study[30]. Another publication, including a larger number of patients, merged patients from different studies to optimize existing NTCP parameters, but bears limitations e.g. of slightly to noticeably varying radiation planning between the included studies and usage of vague data on doses for organs at risk[23]. Results of our study encourage for further medical physics research of the Lyman-Kutcher-Burman (LKB) model parameters as an additional value concerning future toxicities, in particular for currently lower radiation doses. This issue is also addressed in our Response 1 above.
We added a more detailed description in the conclusions-section, as suggested.
The modified passages and the passages answering the reviewer’s comments are highlighted in the manuscript. We hope that our revised manuscript is now suitable for publication in Cancers.
Reviewer 2 Report
The authors present a plan evaluation study regarding dose burden in OARs decreasing PTV volume vs. prescription dose in patients with marginal zone lymphoma
18 patients: 4 plans were generated for each (40 Gy-involved field, 40 Gy-involved site, 30 Gy-involved field, 30 Gy-involved site)
Introduction:
It is a well structured introduction which summarized different dose und target volume concepts in gatric MALT and the aim using Lyman-Kutcher-Burman (LKB) model. It offers an additional value concerning future toxicities by including the volume effects and intrinsic radiosensitivity of organs just as the better assessment of low grade toxicity in abdominal organs
Material and methods:
Which radiation planning tecniques were used for the original plans? How were the CT scans performed? With full or empty bowel, oral contrast, … please describe in 1-2 sentenses.
Follow up was performed, how was the interval to detect early toxicity?
In the following part planning process, evaluation of dose burden and NTCP calculation is explained.
Results:
The autors show the clinical results very detailed and also the results of the plan - compairing part.
First of all median doses were compared in all groups an well described, NTCP s were estimated. Regarding Table 3 +5 it would be easier for the reader if "m" and "n" is described in the footnote.
Discussion:
results and also the limitation of the study were discussed, if have nothing to add.
Author Response
Response to Reviewer 2 Comments
Dear reviewer,
thank you for your reviewing of this manuscript and your opinion on our paper.
Point 1:
Material and methods:
Which radiation planning tecniques were used for the original plans? How were the CT scans performed? With full or empty bowel, oral contrast, … please describe in 1-2 sentenses.
Follow up was performed, how was the interval to detect early toxicity?
Response 1:
As suggested, the following details were added in the ‘Materials and Methods’-section:
Intensity Modulation Radiation Techniques (IMRT) were used for the original plans. The CT scans were performed with an empty stomach after a fast of at least 4 hours or overnight, patients in supine position with arms up using customized immobilization device. A small volume (<50 ml) of oral contrast medium was used in all cases, intravenous contrast medium was recommended if there were suggestive lymph nodes. Respiratory motion has been assessed with fluoroscopy or 4D CT.
According to the reviewer’s comment the interval definition for early toxicity was added in the ‘Materials and Methods’-section:
The interval to detect early toxicity ranged from the therapy start to 90 days after radiotherapy, toxicities occurring later than 90 days after radiation treatment were defined as late toxicities.
Point 2:
Results:
Regarding Table 3 +5 it would be easier for the reader if "m" and "n" is described in the footnote.
Response 2:
To simplify the Table 3 and Table 5 for the reader the three parameters of the LKB model „TD50“, „m“ and „n“ are now described in the footnotes below Table 3 and below Table 5, as suggested.
The modified passages and the passages answering the reviewer’s comments are highlighted in the manuscript. We hope that our revised manuscript is now suitable for publication in Cancers.
Reviewer 3 Report
need more data to confirm the treatment value
This is not conventional treatment modality for the gastric lymphoma and many parameters are not sure .major reset is needed.Author Response
Response to Reviewer 3 Comments
Dear reviewer,
thank you for your reviewing of this manuscript and your opinion on our paper.
Point 1: need more data to confirm the treatment value
This is not conventional treatment modality for the gastric lymphoma and many parameters are not sure .major reset is needed.
Response 1:
We understand the reviewer’s thoughts concerning the quantity of data. According to the reviewer’s comment we have added a detailed description of the involved lymphoma entity, of its prognosis and the prospective work of the study to the introduction-section as well as to the abstract and the conclusions-section:
Indolent lymphomas, in particular of the stomach, are characterized by a low incidence (4 % of gastric neoplasias, 8 % of non-Hodgkin lymphoma), therefore the expected number of cases per year is low (1-2 per 100000). Their prognosis is excellent, patients with gastric marginal zone lymphoma are mostly cured, even by applying progressively lower radiation doses and smaller radiation fields.
For this study as a prospective work, 72 radiation plans with different radiation doses and target volumes (40 Gy IF, 40 Gy IS, 30 Gy IF, 30 Gy IS) were generated for the cohort of 18 patients.
For the high curative purpose of treatment, the potential toxicity of radiation treatment of gastric indolent lymphoma is of rising importance. In contrast, the models and their parameters for calculating normal tissue complication probability (NTCP) of radiotherapy are antiquated, particularly in regard to lymphoma they need to be adjusted to the nowadays lower radiation doses. The findings of this study can be a basis for prospective adaptation of the LKB model parameters, in particular for modern low-dose radiation treatment of gastric lymphoma.
The reviewer is concerned about the conventional treatment modality for gastric lymphoma. We hereby note our description in the introduction-section of the development of conventional radiation treatment modality for the gastric lymphoma regarding radiation doses and field sizes/target volumes. According to the reviewer’s comment we added the description of the modern radiation concept of ISRT with 30 Gy for gastric lymphoma as usual treatment outside of studies in the introduction-section. The study provides deeper insight about the influence of dose versus field reduction in abdominal radiation and besides, and for the first time develops new LKB model parameters for normal tissue complication probability in planning of low dose radiation of gastric lymphoma.
According to the reviewer’s comment that many parameters are not sure, we have added the following descriptions of previous therapy, endoscopic description with follow-up and incidence of helicobacter pylori infection in the results-section:
At diagnosis 6 patients showed helicobacter pylori infection whereas 12 patients primarily were helicobacter pylori negative. Apart from eradication therapy no other previous therapies were applied. At the beginning of radiation therapy all of the 18 patients were helicobacter pylori negative while lymphoma was persisting for at least 12 months after eradication. The endoscopic description before starting the radiation in the majority of patients showed ulcerative mucosal changes as typical pathological finding of the lymphoma, only 2 patients had no adverse mucosal findings. The endoscopic follow-up after radiation treatment had no reports of recurrences, in half of the patients remained a discrete scar or slightly reddened zone at the lymphoma site concerned.
-------------------------------------------------------------------------------------------------------------
|
|
|
|
|
|
|
||
|
|
Responses relating to general Reviewer 3 questions:
These issues were also addressed by Reviewer 1.
Question: Does the introduction provide sufficient background and include all relevant references? Response: We understand the reviewer’s thought on the introduction-section. We hereby note that the introduction is clearly structured and summarizes different dose and target volume concepts in gastric marginal zone lymphoma and the aim of applying the model of Lyman-Kutcher-Burman (LKB). According to the reviewer’s comment we have added a detailed description of the involved lymphoma entity, of its prognosis and the prospective work of the study in the introduction-section. This issue is also addressed in our Response 1 above.
Question: Is the research design appropriate? Response: We thank the reviewer for this comment and hereby note that the LKB model provides additional benefit for estimation of future toxicities, by including volume effects and intrinsic radiosensitivity of organs, allowing a better assessment of low grade toxicity in abdominal organs. This issue is also addressed in our Response 1 above.
Question: Are the methods adequately described? Response: According to the reviewer’s comment more details of the methods were added in the ‘Materials and Methods’-section): Intensity Modulation Radiation Techniques (IMRT) were used for the original plans. The CT scans were performed with an empty stomach after a fast of at least 4 hours or overnight, patients in supine position with arms up using customized immobilization device. A small volume (<50 ml) of oral contrast medium was used in all cases, intravenous contrast medium was recommended if there were suggestive lymph nodes. Respiratory motion has been assessed with fluoroscopy or 4D CT.
The interval definition for early toxicity was added in the ‘Materials and Methods’-section: The interval to detect early toxicity ranged from the therapy start to 90 days after radiotherapy, toxicities occurring later than 90 days after radiation treatment were defined as late toxicities. And in the following part of the ‘Materials and Methods’-section we described the planning process, the evaluation of radiation dose burden and explained the NTCP calculation.
Question: Are the results clearly presented? Response: According to the reviewer’s comment we described very detailed the results with regard to clinical results and to plan-compairing results. We added more data to describe the patients in the results-section: At diagnosis 6 patients showed helicobacter pylori infection whereas 12 patients primarily were helicobacter pylori negative. Apart from eradication therapy no other previous therapies were applied. At the beginning of radiation therapy all of the 18 patients were helicobacter pylori negative while lymphoma was persisting for at least 12 months after eradication. The endoscopic description before starting the radiation in the majority of patients showed ulcerative mucosal changes as typical pathological finding of the lymphoma, only 2 patients had no adverse mucosal findings. The endoscopic follow-up after radiation treatment had no reports of recurrences, in half of the patients remained a discrete scar or slightly reddened zone at the lymphoma site concerned.
We compared and described median doses in all groups and calculated the NTCPs. To simplify the Table 3 and Table 5 for the reader the three parameters of the LKB model „TD50“, „m“ and „n“ are now described in the footnotes below Table 3 and below Table 5, as suggested.
Question: Are the conclusions supported by the results? Response: Thank you for your review. We hereby refer to our description of the results and the discussion of the limitations of the study in the discussion-section. First of all, the results allow limited conclusions due to the involved number of n=18 patients. But since this study involves a rare disease, the depicted 3 parameters showing significant differences in all compared subgroups of the 72 radiation plans represent pioneering results. According to the reviewer’s comment we added comparable studies of other authors with similar numbers of patients in the discussion-section: Published studies on NTCP for other organs at risk after radiation treatment comprise various patient numbers and provide different informative value. Other authors using a similar approach by generating comparable radiation plans with consistent radiation dosage reported about similar limited numbers of patients as involved in this study[30]. Another publication, including a larger number of patients, merged patients from different studies to optimize existing NTCP parameters, but bears limitations e.g. of slightly to noticeably varying radiation planning between the included studies and usage of vague data on doses for organs at risk[23]. Results of our study encourage for further medical physics research of the Lyman-Kutcher-Burman (LKB) model parameters as an additional value concerning future toxicities, in particular for currently lower radiation doses. We added a more detailed description in the conclusions-section, as suggested.
The modified passages and the passages answering the reviewer’s comments are highlighted in the manuscript. We hope that our revised manuscript is now suitable for publication in Cancers.
|
||||||
Round 2
Reviewer 1 Report
I have read the second version. It's now acceptable for publication.
Reviewer 3 Report
more appropriate after revision